# Photolyase Production and Current Applications: A Review

**DOI:** 10.3390/molecules27185998

**Published:** 2022-09-15

**Authors:** Diana Ramírez-Gamboa, Ana Laura Díaz-Zamorano, Edgar Ricardo Meléndez-Sánchez, Humberto Reyes-Pardo, Karen Rocio Villaseñor-Zepeda, Miguel E. López-Arellanes, Juan Eduardo Sosa-Hernández, Karina G. Coronado-Apodaca, Ana Gámez-Méndez, Samson Afewerki, Hafiz M. N. Iqbal, Roberto Parra-Saldivar, Manuel Martínez-Ruiz

**Affiliations:** 1Tecnologico de Monterrey, School of Engineering and Sciences, Monterrey 64849, Mexico; 2Tecnologico de Monterrey, Institute of Advanced Materials for Sustainable Manufacturing, Monterrey 64849, Mexico; 3Department of Basic Sciences, Universidad de Monterrey, Av. Ignacio Morones Prieto 4500 Pte, San Pedro Garza Garcia 66238, Mexico; 4Koch Institute for Integrative Cancer Research, Massachusetts Institute of Technology, Cambridge, MA 02142, USA; 5Division of Gastroenterology, Brigham and Women’s Hospital, Harvard Medical School, Boston, MA 02115, USA

**Keywords:** UV radiation, UV damage, enzyme, photolyase immobilization, bioactive compounds

## Abstract

The photolyase family consists of flavoproteins with enzyme activity able to repair ultraviolet light radiation damage by photoreactivation. DNA damage by the formation of a cyclobutane pyrimidine dimer (CPD) and a pyrimidine-pyrimidone (6-4) photoproduct can lead to multiple affections such as cellular apoptosis and mutagenesis that can evolve into skin cancer. The development of integrated applications to prevent the negative effects of prolonged sunlight exposure, usually during outdoor activities, is imperative. This study presents the functions, characteristics, and types of photolyases, their therapeutic and cosmetic applications, and additionally explores some photolyase-producing microorganisms and drug delivery systems.

## 1. Introduction

In the late 1940s, Albert Kelner reported the phenomenon of enzymatic photoreactivation for the first time, studying in situ the repair of major DNA lesions produced by ultraviolet (UV) radiation by a light-induced enzymatic cleavage of a thymine dimer to yield two thymine monomers. This enzyme is called photolyase, and it detects and binds to dimers contained in single- and double-stranded DNA [1]. Photolyase is also active against cytosine dimers and cytosine–thymine dimers, which are also formed by UV irradiation, but with much less frequently.

Photolyases are evolutionary ancient flavoproteins found in archaea, bacteria, and eukarya domains [2,3]. In vertebrates, photolyases are found in fish, amphibians, birds, and reptiles. Interestingly, mammals have lost this mechanism of protection throughout evolution. Hence, most mammals, including humans, can only repair these DNA lesions through a process called nucleotide excision repair. UV light is a known source of damage to our DNA that can be repaired by this mechanism. However, the deficient repair of UV-induced DNA damage which, for example, may occur after excessive unprotected sunbathing, can cause damage to DNA by inducing the formation of a cyclobutane pyrimidine dimer (CPD) and a pyrimidine-pyrimidone (6-4) photoproduct (6-4PP). Both CPD and 6-4PP can lead to cellular apoptosis and mutagenesis, which can eventually lead to skin cancer.

According to the World Cancer Research Fund International [4], between 2 and 3 million non-melanoma skin cancers and 132,000 melanoma skin cancers are estimated to occur globally each year. Importantly, the main factors that promote the development of melanoma are exposure to the sun and a history of sunburn. Commonly, this chronic, long-term sun exposure causes lesions such as premature aging and actinic keratosis, both associated with skin cancer [5].

The increasing incidence of skin cancer and the consequent burden to the healthcare system poses great problems, since the primary topical product defense has been broad-spectrum (UVA/UVB) sunscreen application. Nonetheless, it has been shown that some sunscreen products contain chemicals that can enter the bloodstream [6], and some products were found to contain cancer-causing substances. In addition to these deleterious effects on health, the environmental impacts of sunscreen ingredients include increasing concerns regarding coral reefs. Therefore, the use of repair DNA enzyme-based products represents a feasible alternative to prevent and treat DNA damage and consequent skin lesions.

This review focuses on the characterization of photolyases and therapeutic and cosmetic applications available.

## 2. Photolyase

Photolyase is a protein that has various functions, among which is the repair of DNA damaged from exposure to UV rays from the sun [7].

Additionally, the presence of photolyases has been observed in fish, amphibians, birds, and a few marsupials [8]; nevertheless, in higher plants and animals, the ability to repair DNA was lost during evolution (Figure 1). Therefore, their function is limited to regulating growth and acting as blue-light photoreceptors; these enzymes are known as cryptochromes [9].

The first precedent of enzymes with photolyase-like activity was discovered in 1993 in a plant of the *Brassicaceae* family native to Europe, *Arabidopsis thaliana* [10]. In other plants such as white mustard (*Sinapis alba*), the same photolyase cofactors are present; however, these plants lack DNA repair activity [11]. However, the photoreactivation process and DNA repair with photolyase are available and have been demonstrated in *Streptomyces griseus* [12] and bacteriophages [13].

In cyanobacteria, the existence of photolyase is reported in *Anacystis nidulans* [14]. The activity of photolyase enzymes has been reported mainly in environments with high exposure to UV rays, such as the case of twelve species of diatoms from Antarctica that demonstrated a DNA repair response to UV radiation damage [15]. 

In eukaryotic organisms, the Antarctic alga *Chlamydomonas* sp. *ICE-L*, which is developed in high-irradiation environments, can use mechanisms that reduce UV radiation damage [16].

Figure 2 shows the evolutionary relationship of several members of the photolyase superfamily. From the members analyzed, the phylogenetic tree highlights two main groups: one dominated by photolyases from microalgae and plants, and the other with more diversified evolutionary relationships. Interestingly, some microalgae species express photolyase at a relatively near distance to human photolyases.

### 2.1. Type of Enzyme

Photolyases are monomeric proteins with a molecular mass from 50 to 61 kDa. They are made up of 450–550 amino acids and two unsorted covalently bound chromophores as cofactors. One of the cofactors is always flavin adenine dinucleotide FAD, and the second is methenyltetrahydrofolate (MTHF) or 8-hydroxy-7, 8-didemethyl-5-deazariboflavin (8-HDF) [20]. Light is an indispensable resource in the photoreactivation process for the conversion of enzyme–substrate complexes into additional DNA repair products [21]. The surface of photolyases is characterized by a positive charge near the substrate binding which promotes the interaction with DNA [22]. The structure of photolyase consists of two domains: a C-terminal α-helical catalytic domain that contains the flavin cofactor and an N-terminal α/β domain [23].

The superfamily of chromophores/photolyases (CRY/PHR) consists of subfamilies (Figure 3) [24,25], of which there are three types of photolyases that have been identified to repair a specific type of dimer: (1) CPD photolyase, responsible for repairing CPD, (6-4); (2) photolyase, which repairs (6-4) pyrimidine pyrimidone; and (3) cryptochrome-DASH, which causes a variety of physiological changes to DNA [2,26].

### 2.2. Photolyase and Microorganisms

Photolyase biosynthesis begins with the transfer of an electron from the anionic chromophore FADH^−^, promoted by light. In a catalytically active form, this binds to damaged DNA in a high-affinity, light-independent step [27,28].

To study numerous characteristics of the photolyases obtained from various organisms such as structural, physical, and mechanical properties, researchers have employed genetic manipulation approaches to promote the overexpression of DNA repair enzymes, since biosynthesis naturally provides low concentrations. Subsequently, extraction and purification processes are performed to obtain better quality enzymes (Table 1) [29]. After these steps, generally, between 15 and 25 mg of photolyase with a purity greater than 98% is obtained. The quality of the enzyme obtained is reflected in the color of the extract: a dark blue color indicates a good quality product. Therefore, it is important to measure and quantify the absorbance spectrum after enzyme purification [29].

### 2.3. DNA Damage by UV Irradiation

The organisms and cells that inhabit the earth naturally are continuously exposed to genotoxic agents present in the environment. Sunlight, as a source of UV rays, is one of the main genotoxic agents; nonetheless, it is essential for the development of life, for example, in the process of photosynthesis [40].

The damage caused by exposure to UV radiation (Figure 4) can trigger various skin reactions, mainly (as mentioned before) by affecting pyrimidine dimers; erythema, immunosuppression, and melanogenesis are just some of the disorders that can occur. Hence, it has been proven that overexposure to sunlight almost irreversibly damages skin cells [41].

### 2.4. Photolyase Mechanism of Action

There are multiple mechanisms of DNA repair: direct reversal [43], base excision, nucleotide excision [44], mismatch [45], single-strand break, and double-strand break repair [46].

Photolyases are light-driven DNA repair enzymes which function specifically in the reversal of genomic lesions induced by UV radiation [47]. An important DNA repair mechanism for mutagenic and cytotoxic UV-induced photolesions in DNA is photoreactivation, which utilizes enzyme photolyase for reverting modified nitrogenous bases into normal form, employing blue wavelength [48]. DNA repair to minimize mutagenic changes is divided into two main mechanisms: single-strand (ss) and double-strand (ds) DNA damage repair. In the same way, ss and ds DNA repair are divided into direct reversal repair, nucleotide excision repair, base excision repair, and mismatch repair for ss DNA repair and homologous recombination and non-homologous end-joining repairs for ds DNA repairs [7].

The photolyase for the CPD and 6-4PP lesions can be divided into CPD photolyases based on the photoproduct that they recognize, which are subdivided in relation to the amino acid sequence they possess, and 6-4PP photolyases, respectively [49]. Photolyases can possess flavin adenine dinucleotide (FAD) in four different redox states: oxidized (FAD), anionic semiquinone (FAD^−^), neutral semiquinone (FADH), and anionic hydroquinone (FADH^−^) [7]. Different redox states will act differently in the absorption spectrum. While FAD and FAD^−^ mainly absorb UV-A, and blue light, FADH absorbs blue, green, and red light, and FADH^−^ does not absorb visible light. An absorption spectrum was determined from two organisms by following flavin intermediates during the catalytic process, showing the mode of action on the DNA repair by photolyase based on the absorption properties of FAD [50].

According to Wang et al. [50], DNA repair by photolyase can be divided into three steps (Figure 5): (i) Recognition, which is a light-independent process where CPD or (6-4) photoproduct in the damaged DNA forms a photolyase/DNA complex by flipping into the active site containing the flavin cofactor of the DNA photolyase. (ii) During the catalytic reaction [51] taking place when FAD is at a fully reduced state (FADH^−^), the methenytetrahydrofolate (MTHF), a photolyase chromophore, transfers energy to FADH^−^ through the absorption of photons from the blue-light spectrum, changing FADH^−^ to an exciting form of FADH^−^. In this step, CPD or the (6-4) photoproduct ring is opened by bond dissociation in the dimer radical anion, and electron transfers from FADH^−^ to the lesions occur [7,52]. (iii) Finally, the separation step completes the repair process by the departure of repaired DNA from photolyases.

As previously mentioned, FAD oxidation states influence the light absorption spectra; hence, catalytic reactions for DNA repair from FADH^−^ take place under blue-light conditions. Considering the poor permeability of living tissues to blue light, works that aim to remove the barrier of this range of light have been developed using harvesting or antenna chromophores. Antenna chromophores are utilized by some photolyase to gain the photoreception ability of a light range, and the development and application of artificial antenna chromophores demonstrate an increase of up to 1.5-fold in DNA repair activity. This is a promising strategy area for the optimization of photolyase DNA repair [52]. Nevertheless, reports on the application of artificial antenna chromophores remain scarce.

### 2.5. Immobilization and Biocarriers

Photolyase offers outstanding protection and recovery mechanisms against sunlight-induced DNA damage, but if the enzyme lacks the capacity of reaching the site where it should intervene, or lacks the stability necessary to complete this, its utilization will go to waste.

Currently, immobilization strategies such as the encapsulation of photolyases within liposomes are the most used method to stabilize and deliver the protein; nonetheless, other methods, such as the application of nanomaterials, are being explored [49]. There are an assortment of technologies and systems currently being used as an approach to deliver active ingredients in skincare, such as antioxidants that can be useful and might offer an alternative to liposomes. Some of these technologies include the use of different formulations, such as gels, hydrogels, and emulsions (micro and nano-emulsions), the usage of other vesicular delivery systems instead of liposomes, such as ethosomes, transfersomes, niosomes, and non-vesicular particles such as solid lipid nanoparticles, and nanostructured lipid carriers, polymeric nanoparticles, nanocrystals, and—lastly—carbon, metal, or metal oxide nanoparticles. These immobilization technologies seek to improve long-term stability and sensitivity in order to provide a better catalytic activity; however, it is necessary to explore their efficiency according to their final applications [54,55,56,57,58,59,60,61].

## 3. Photolyase Applications

The sun is the most vital star in our solar system. The radiation from this star promotes photosynthesis in photoautotroph organisms, and in mammalian species, light promotes the synthesis of vitamins necessary for the bone structure, such as the production of vitamin D, among other things [62,63]. However, everything in excess can be harmful, and high exposure to UV rays can be harmful to cells [64,65]. For this reason, some organisms, such as bacteria and algae, which are constantly exposed to UV radiation, have evolved to develop several mechanisms that prevent or even revert damages caused by the exposure of different UV wavelengths. Unfortunately, most organisms, humans included, do not have the mechanisms to mitigate the effects of prolonged exposure to UV rays, with the consequence of causing damage to DNA which ultimately can result in the development of skin diseases, or even skin cancer.

The use of enzymes that promote DNA repair, such as photolyase, represents a key alternative for a wide range of applications in skin care products aimed to prevent or reverse the damage prompted by extensive sunlight exposure. Advances in immobilization within liposomes and nanomaterials have made the application of photolyase more accessible in a wide range of commercial products, due to its higher stability. The activity of the enzyme is maintained even in harsh conditions [49]. Photolyase is typically used in topical creams or sunscreens which are used to prevent UV damage, and in therapy to restore the skin from conditions such as premature photoaging, actinic keratosis, and squamous cell carcinoma. 

In the following sections, we will discuss the different applications of photolyases in different fields and cosmetics, and explore some examples of them being used and their effects on different organisms or skin conditions. Table 2 presents some studies related to the use of sunscreens containing photolyase and their effects on the treatment of premature photoaging, actinic keratosis, and some types of skin cancer.

### 3.1. Current Photolyase Production

As stated previously in the manuscript, there are many organisms capable of producing photolyase to repair DNA damage from UV exposure, and currently, the industry is focusing on the production of this enzyme to take advantage of its properties to improve crops and its potential use in therapeutic products [28].

The current application of the photolyase enzyme is focused on the enhancement of UV resistance not only in microorganisms, but also in other organisms and areas. An example of photolyase application is the use of it to improve agriculture practices, which has been reported to improve plants. For example, in Japan, researchers successfully modified African rice to overexpress the gene. CPD repaired enzyme photolyase, demonstrating an increased UVB resistance compared to the rice plants without these modifications [66]. Similarly, there is open research about improving the UV irradiation resistance of fungal insecticides to improve their pest control [67].

### 3.2. Sunscreen with Photolyase as an Ingredient

The use of products to protect the skin from the sun is something humankind has undertaken for centuries, from the Egyptians’ use of different plant extracts to the sunscreens we know today, composed of a wide variety of filters offering protection beyond UV radiation [68,69]. In the last few decades, the use of sunscreens has been promoted and advertised to prevent damage to the skin by UV radiation; thus, the sunscreen market is expected to reach USD 24.4 billion by 2029 [68,70].

Despite the proven advantages of regular sunscreen application (from reducing the effects of photoaging to protecting against skin cancer) [71,72], there are some concerns about the systematic and regular use of sunscreens and the effect on vitamin D synthesis, especially in older people; nonetheless, there is plenty of evidence that deny those claims and assure that the usage of sunscreens does not affect levels of vitamin D [73,74,75]. Another concern with sunscreens is the skin absorption of the most used active ingredients, such as avobenzone and oxybenzone, among others, resulting in high plasmatic levels [5,6]. In addition to these health concerns, the damage these organic ingredients cause to the environment has been studied [76,77]. However, these findings prove the importance and the need to further assess the products used in sunscreens, and to find active ingredients that can offer the same benefits for all ages and skin types without a detrimental effect on human and environmental health [78].

The use of the photolyase has been around for the last few decades, with the first patent in 1988 [79], and since then it has been tested and proven that this enzyme can prevent and reverse sunlight-induced skin damage when used as an active ingredient in traditional sunscreen formulations [7,80,81,82]. 

There are sunscreens containing DNA repair enzymes obtained from microalgae and labeled as “plankton extract” available on the market. Some companies that offer this product are Isdin (Barcelona, Spain) with the product Eryfotona^®^ AK-NMSC, Kwizda Pharma GmbH (Vienna, Austria) with Ateia^®^, and Pharma Cosmetics (Oradell, NJ, USA), with a variety of products such as Neova Active^®^, Neova Everyday^®^, and Neova Silc Sheer^®^ 2.0. Many other companies are also releasing new products containing photolyase [82]. For a more in-depth listing of all the products currently available on the market, we recommend consulting the Supplementary Material of the review by Yarosh et al. [83].

### 3.3. Photoaging

Normal signs of aging are the appearance of fine lines, pigmentation, and wrinkles in the skin, all of which appear over time and are generally attributed to getting older; however, there is evidence that multiple factors contribute to accelerating the aging process, including lifestyle factors such as smoking, sleep, diet, the chronic use of drugs, environmental factors such as contact with polluted air, exposure to visible and infrared light [84], and UV radiation, with the last one accounting for up to 90% of visible changes on the skin [85,86].

Photoaging is a term that has been used since 1986 [87] to reference the effect on the skin produced by chronic exposure to UV radiation that causes damage to DNA [82,84,88,89] and leads to premature aging. All of these signs overlap with natural signs of aging, such as wrinkles, thin and dry skin due to a loss of underlying fat, more fragile skin, and pigmentation; however, the effects of photoaging go beyond these appearances with, depending on the skin type, the formation of fine to coarse wrinkles, a leather-like appearance to the skin, and hyperpigmentation or dyschromia. All of these signs are presented even when normal signs of aging have not appeared [85,90,91,92].

Given that photoaging only occurs in areas of the skin that have been exposed to the sun for a prolonged time, it is recommended to avoid sun exposure and apply sunscreens that provide protection against UVA and UVB radiation and can even in some cases reverse damage [71,85,93]. Listed below are some studies related to the use of photolyase in clinical trials intended to assess the efficacy of this enzyme to mitigate the effects of photoaging.

Corinne Granger et al. conducted a study in which they tested, over 28 days, a tinted sunscreen containing encapsulated photolyase on 30 women of ages ranging from 45 to 65 with slight-to-moderate photoaging signs [94]. The patients were evaluated on day zero and on the last day of use and signs of photoaging were analyzed and evaluated. The results showed an improvement between 6 and 12% for each factor analyzed in the treated women compared to the control.

Another study was conducted to prove the efficacy of using photolyase from the microalgae *Anacystis nidulans* to prevent damage to DNA produced by UV radiation on the skin. This was performed by comparing the use of a regular sunscreen with a sun protection factor (SPF) 50, and the same sunscreen supplemented with photolyase [95].

The results from this study revealed that, for the 10 participants (5 males and 5 females, with ages ranging from 26 to 36 years old and Fitzpatrick skin type II), the formation of CPDs and apoptosis of the skin cells was reduced by 93%, and 82%, respectively, with the use of the sunscreen containing photolyase compared with the control that received only radiation.

Similarly, Emanuele et al. [96] compared a novel topical product containing a traditional sunscreen with SPF 50, a mixture of DNA repair enzymes encapsulated in liposomes, one of them being photolyase, and an antioxidant complex versus other topical products with similar characteristics. The study demonstrated the novel topical products as the most effective in reducing the three parameters analyzed: the formation of CPD, 8-oxo-7,8-dihydro-2′-deoxyguanosine (8OHdG), and protein carbonylation (PC). This was attributed to the synergistic effect of all the components within the product.

**Table 2 molecules-27-05998-t002:** Studies on the use of photolyase in different skin conditions.

Study	Skin Condition	Therapy Evaluation	Duration	Assessment	Results	Reference
Clinical series	AK	Eryfotona	3 months	Clinical photography	Great improvement in AK lesions count.	[97]
Retrospective case study	Xeroderma Pigmentosum	Eryfotona	12 months	Histological records	Reduction of 65% for AK, 56% for BCC, and 100% for SCC lesions.	[98]
Longitudinal, observational clinical	AK	Eryfotona	3 months	Clinical, dermoscopy, and confocal microscopy analysis	Grade I AK clinical and dermoscopy improvement. Reduction in desquamation. Improvement in the epidermal architectural pattern. Grade II AK, no improvements.	[99]
Pilot study	AK	Eryfotona	1 month	Clinical, dermoscopy, and reflectance confocal microscopy assessments	Erythema and scaling improvement.	[100]
Clinical	AK	Eryfotona	9 months	Telethermografy	Hyperthermic halos area reduced from 3.46 to 0.64 cm^2^.	[101]
Randomized, assessor-blinded parallel-group	AK	Eryfotona	9 months	Lesion count	Significant reduction in new AK lesions. No additional photodynamic therapy required.	[102]
Prospective observational study	AK	Eryfotona Cryotherapy	6 months	Epidemiologic, clinical, and therapeutic variables	No adverse cutaneous effects and 84% improvement in AK lesion count.	[103]
Prospective, single-arm, case-series	AK	Eryfotona	3 months	Clinical photography	Partial response in 100% of patients. 50% reduction in lesion count.	[104]
Randomized, double-blind parallel-group Pilot study	AK	Eryfotona	6 months	Clinical, dermoscopy, and reflectance confocal Microscopy evaluation	Significant reduction in mean AK lesion number up to 31%.	[105]
--	Photoaging	Tinted facial sunscreen with high sun protection, peptide complex, and encapsulated photolyase	1 month	Periocular wrinkles, skin firmness and elasticity, UV spots, and patient subjective questionnaire	Wrinkle count −6.9%. Wrinkle volume −10.4%. UV spots area −9%. Firmness +8.2. Elasticity +11.3%.	[94]
--	UV exposure	Sunscreen amended with photolyase	4 days	Skin biopsies after experimental irradiations	93% prevention of CPD formation. 82% apoptosis prevention.	[95]
Head-to-head comparison studies	UV exposure	Triple-protection factor broad-spectrum sunscreen (TPF50)	--	Skin biopsies after experimental irradiations	Reduction in CPD and protein carboxylation.	[96]
Randomized, double-blind, factorial clinical trial	AK	Sunscreen amended with photolyase	2 months	Clinical and demographic variables	No significant differences with common sunscreen.	[106]
--	AK	Eryfotona	1 month	Histopathological and molecular assessment	Improvement in the field of cancerization. Restoration of normal phenotype through CPI-17 up-regulation.	[107]
Randomized, clinical study	AK	Sunscreen amended with photolyase	6 months	Fluorescence diagnostics using methylaminolevulinate Skin biopsies	Superior to sunscreen in reduction in field cancerization and UVR-associated molecular signatures.	[108]

AK: actinic keratosis. SCC: squamous cell carcinoma. BCC: basal cell carcinoma. CPD: cyclobutane pyrimidine dimer.

### 3.4. Actinic Keratosis

Actinic keratosis (AK) is a skin disease that is characterized by squamous lesions that histologically show keratinocyte neoplasms occurring on skin that has had long-term exposure to UV radiation [109]. AK is typically presented in people with light skin belonging to the Fitzpatrick skin types of I to III in areas of the skin on which they experience solar exposure regularly, such as the head (especially in areas with hair loss), ears, neck, forearm, and the dorsum of the hand [80,110]. The lesions of AK can progress to keratinocyte carcinoma, the most common type of skin cancer in the United States [111]. Therefore, AK must be prevented or treated early to avoid further disease progression. The method to prevent skin damage is to avoid sun exposure, but when the exposure is inevitable it is recommended to use protective clothes and at least 2 mg per cm^2^ of sunscreen in the exposed areas [112]. However, these measures are not enough once the skin damage is present and the use of active molecules capable of stopping and even reversing this damage is essential [113].

Some studies have been published with diverse findings. For example, the efficacy of photolyase in sunscreen and a combination of topical antioxidants in the treatment of patients with AK were assessed in Brazil [106]. A total of 80 patient forearms were tested using either regular sunscreen or sunscreen containing photolyase during the day and the night. They either applied topical antioxidant or placebo cream to one forearm for 8 weeks. The researchers found that all groups tested showed a significant improvement at the end of the study; however, there were little to no significant differences between the groups using regular sunscreen and those with additional photolyase. The authors attributed this to the short time period of the treatment.

Puig-Butillé et al. [107] evaluated the use of a film-forming medical device containing photolyase in liposomes on a small group of patients from a wide range of ages, composed mainly of males that presented multiple AK and two patients with xeroderma pigmentosum. The study was conducted for 4 weeks, and at the end of the study, all groups showed an improvement in their condition. Notably, some showed total clearance on the assessed area from lesions caused by UV radiations. With the same medical device with photolyase, Eibenschutz et al. [102] analyzed the effect of the product with the enzyme compared to a regular sunscreen on 30 patients that underwent photodynamic therapy (PDT) with a total of 225 AK lesions for 9 months. At the end of the treatment, it was shown that the group treated with the film-forming medical device did better than the group treated with regular sunscreen. PDT was not needed, nor was any other medical procedure, and no new AK lesions were observed.

A research study in 2015 compared the effects of sunscreen containing photolyase and traditional sunscreens in 28 patients during 6 months of treatment [108]. The findings showed that both treatments reduced hyperkeratosis; however, for the field cancerization and the levels of CPDs, the results showed a better performance in the group that used the sunscreen containing photolyase than the group that used the traditional sunscreens.

### 3.5. Skin Cancer

Skin cancer remains a major global public health threat [114]. As the human body naturally grows, cells are divided when needed and die when they lose their normal function, or due to natural cell-aging. Cancer starts as a result of an interference in the cycle of cell growth division and death. This condition is characterized by an overproduction of cell division and the permanency of abnormal cells, instead of their death [115].

According to the American Cancer Society, there are five types of skin cancer: basal and squamous cell skin cancer, melanoma skin cancer, Merkel cell skin cancer, lymphoma of the skin, and Kaposi sarcoma. The basal and squamous cell skin cancers are mostly found on the body areas commonly exposed to the sun without protection, such as the head, neck, and arms. These two types are the most common, and they start in the epidermis [115]. In early-stage cases, a skin excision is the treatment for squamous cell carcinoma (SCC) [116].

The precursor cell of SCC is AK, and for BCC it is hypothesized that its occurrence is related to interfollicular epidermal basal keratinocytes with retained basal morphology from the follicular outer root sheath or sebaceous gland-derived keratinocytes [117]. Malignant melanoma is a serious form of skin cancer that begins in cells known as melanocytes. While it is less common than SCC and BCC, melanoma is the most severe type of skin cancer due to its capacity to spread if it is not treated at an early stage [118,119]. Merker cell, lymphoma, and Kaposi cancers are less common types of skin cancer [115].

Field cancerization refers to the replacement of the normal cell population by a cancer-primed cell population that may show no morphological change [101]. Some studies have focused on this topic and the role of photolyase as a potential treatment. A study with a topic product categorized as a medical device containing photolyase showed positive results for treating cancerization areas with long-term use versus the use of commercially available sunscreen, not only in terms of Baseline Severity Index (BSI) and total Clinical Score (TCS), but also by reducing the occurrence of new AK lesions [105].

In 2016, Naverrete-Dechent et al. [104] showed that subjects with skin field cancerization showed a partial positive response to the treatment with a photolyase-added sunscreen and at least a 50% reduction in their lesion number. These findings are consistent with the work of Laino et al. [101], where 30 individuals with AK were treated with a photolyase-added medical device, which improved their lesions.

## 4. Perspectives

Photolyase’s potential for DNA repair has been widely studied in vitro and in vivo. Especially, numerous studies have investigated the impact of the use of products containing DNA repair enzymes in patients. Still, we found that one of the major limitations of these studies is the duration of the treatment, making it difficult to determine the effect of photolyase; thus, several details remain as open questions. In this sense, it would be advisable for researchers to consider longer-term treatments in order to achieve solid information regarding the effect of photolyase in cosmetic use.

On the other hand, the mechanism of photolyase DNA repair has been widely studied, which makes it plausible to begin the evaluation of possible modifications to increase the activation of photolyase, even under unfavorable conditions. The use of artificial antenna chromophores is an area of opportunity that has already shown results, but the literature remains scarce. Additionally, microalgae biomass is a potential source of photolyase enzymes. This represents an alternative source in the application of technologies where microalgae can be implemented for bioremediation processes such as CO_2_ sequestration and water treatment, and the produced microalgae biomass could further be used to extract photolyase. These biomass production processes also represent a low-cost approach for generating the enzyme, as a substrate is not needed during the production of microalgal biomass.

## 5. Conclusions

The exploration of photolyase production improvement can be implemented to take advantage of several microorganisms. Biotechnologies for the development of new products that mitigate the DNA damage produced by UV radiation have been explored. Technological advances have allowed the extraction and use of photolyases in different products. Furthermore, new efforts for immobilization and delivery systems are needed to develop new topical formulations to battle UV-damaged DNA affections.

## Figures and Tables

**Figure 1 molecules-27-05998-f001:**
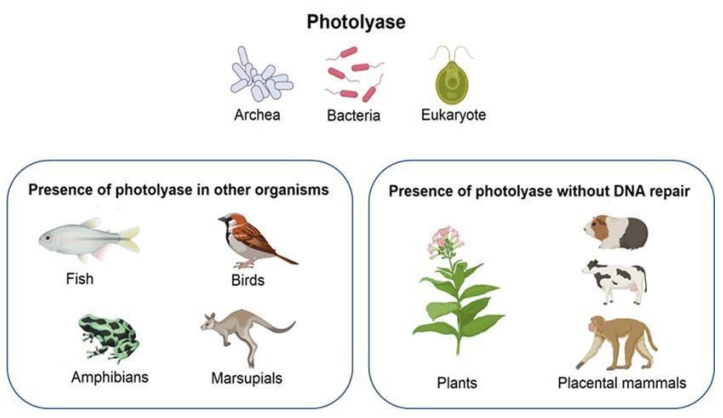
Presence of photolyase in different organisms. Due to evolution, some species lost the ability to repair DNA. Created with BioRender.com.

**Figure 2 molecules-27-05998-f002:**
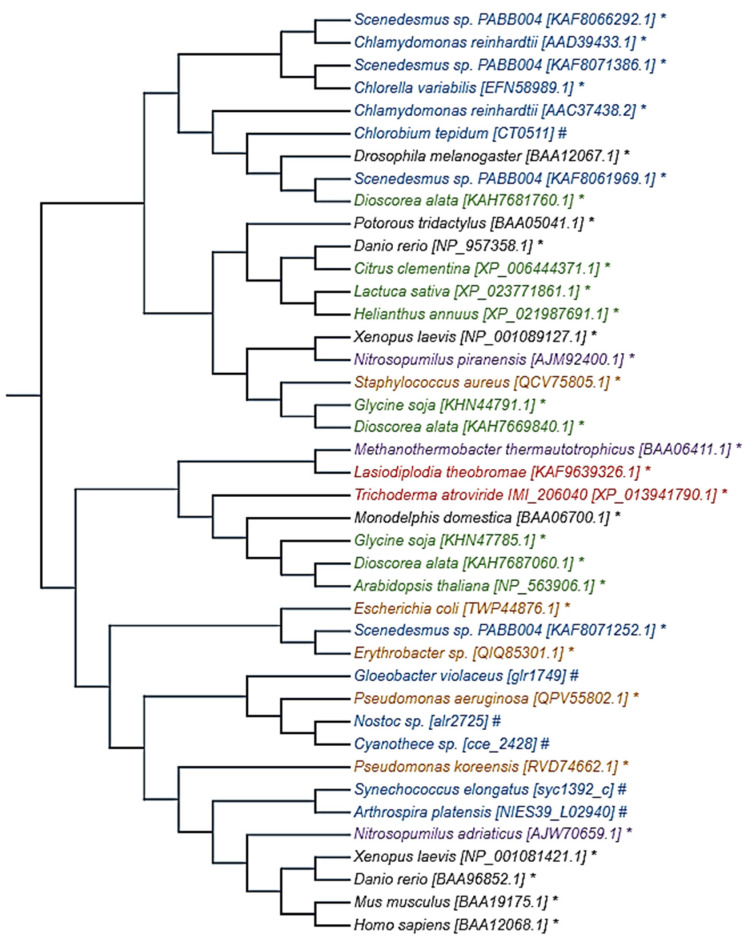
Phylogenetic tree of photolyase. An evolutionary relationship among several members of the photolyase family is shown. This relationship was inferred using the neighbor-joining method [17] with the Poisson correction method [18] in MEGA11 [19]. Amino acid sequences from different species were obtained from the NCBI protein database (*) and CyanoBase database (#), and ID number is between square brackets. Colors are used as group identifiers: black for animals, green for plants, blue for microalgae (eukaryotic and cyanobacteria), red for fungi, orange for bacteria (except for cyanobacteria), and purple for arquea.

**Figure 3 molecules-27-05998-f003:**
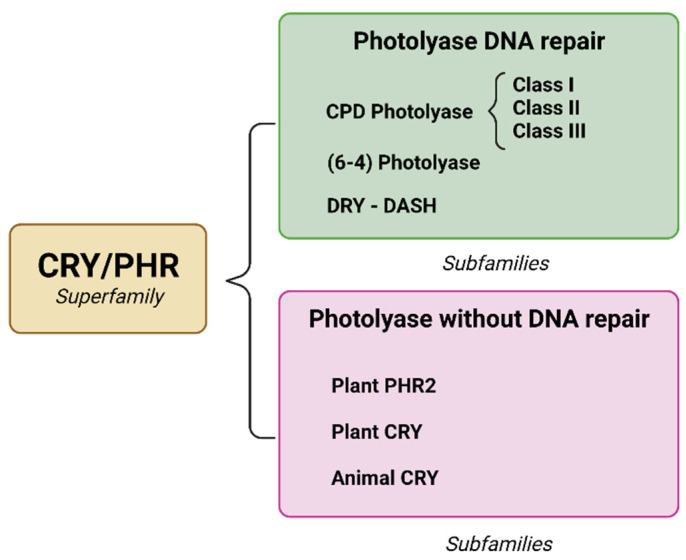
Integration of the CRY/PHR superfamily and the emergence of subfamilies due to evolutionary changes. The main distinctions are the ability to repair specific DNA and loss of photorepair capacity. Created with BioRender.com.

**Figure 4 molecules-27-05998-f004:**
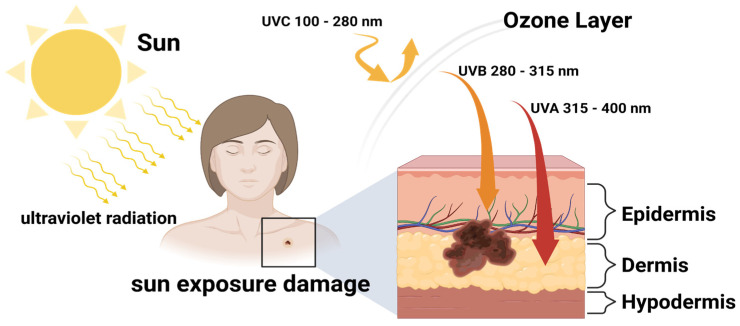
UV light damage to the skin. UV radiation is composed of UVA and UVB rays, and has the ability to penetrate the skin layers. Various UV rays will have different impacts and effects on the skin; UVA and UVB rays are mainly linked to long-term skin damage and play a key role in some skin cancers [42]. Created with BioRender.com.

**Figure 5 molecules-27-05998-f005:**
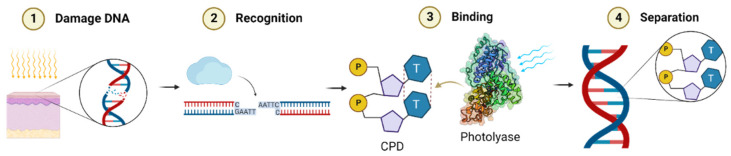
DNA repair by photolyase. DNA damage can be repaired by the enzyme photolyase in three steps: recognition of the damage, binding of the photolyase and the DNA damage, and lastly the separation of the enzyme resulting in the repaired DNA [47,53,50]. Created with BioRender.com.

**Table 1 molecules-27-05998-t001:** Studies demonstrating the presence of different types of photolyases in various organisms and the extraction and purification methods necessary to prove their DNA repair activity.

Microorganisms	Genus	Type Photolyase	Extraction and Purification	References
*Agrobacterium fabrum*	Prokaryote	(6-4) Photolyase	Heated and cleared by centrifugation HPLC column from Macherey and Nagel	[30]
*Rhodococcus* sp. *NJ-530*	Marine bacterium	CPD Class I	Disrupted with ultrasonication, Ni-NTA resin	[31]
*Chlamydomonas* sp. *ICE-L*	Psychrophilic microalga	(6-4) Photolyase	Disrupted with ultrasonication, Ni-NTA resin	[32]
*Hymenobacter* sp.	Antarctic bacterium	CPD Class I	Lysed with sonication, Ni-NTA resin	[33]
*Methanosarcina mazei Mm0852*	Archaea	CPD Class II	Cell disruption with lysozyme, EDTA and PMSF with an emulsifier, Ni-NTA resin	[34]
*Pohlia nutans M211*	Antarctic Moss	CPD Class II and (6-4) Photolyase	Ultrasonic cell disruptor, Ni-NTA resin	[35]
*Phaeodactylum tricornutum ICE-H*	Antarctic diatom	CPD Class II	Ultrasonic cell disruptor, Ni-NTA resin	[36]
*Caulobacter crescentus*	Oligotrophic bacterium	CPD Class III	Heated and cleared with centrifugation, purified by affinity chromatography on amylose resin	[37]
*Mucor circinelloides*	Fungus	CRY-DASH	Disrupted with a French press, affinity chromatography-HisTrap HP column	[38]
*Phycomyces blakesleeanus (NRRL1555)*	Fungus	CRY-DASH	Disrupted with a French press, affinity chromatography-His Trap HP column	[39]

## Data Availability

Not applicable.

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
