# Peer review of "Photolyase Production and Current Applications: A Review"

_molecules, 2022, doi:10.3390/molecules27185998_

Round 1

Reviewer 1 Report

In this review the authors presents photolyases and their functions, characteristics as well as therapeutic and cosmetic applications and this has been presented in a perfect and concise/updated manner. I have only few suggestions, instead of text material i suggest authors to include schematic/pictorial representations of mechanisms and schemes with respect to biomolecules in all aspects in this theme. As well as graphical abstract the entire theme should be included. I am happy to recommend acceptance of this manuscript in Molecules.

Author Response

Dear Reviewer,

Figure 5 were included to attend your comments

Reviewer 2 Report

Till now, many reviews about photolyases have been published, and the catalytic mechanisms of photolyases have been well summarized in detail. This paper reviews the possible applications of photolytic enzymes, which is a novel and meaningful perspective.

I suggest list the types of photolyases used in different sections of “3. Photolyase Applications”, it’s also will be good list the types of photolyases in table 2.

Since this review focused on the application of photolyases, I suggest added one section about the current situation of expression and production of photolyases, which were used in different applications.

Author Response

Dear Reviewer,

The whole manuscript was improved to attend to your recommendations.

Please see the highlighted part in the manuscript

Round 2

Reviewer 1 Report

The authors have incorporated suggested concerns and i have no more comments.